# Measurement of Sea Waves

**DOI:** 10.3390/s22010078

**Published:** 2021-12-23

**Authors:** Giovanni Battista Rossi, Andrea Cannata, Antonio Iengo, Maurizio Migliaccio, Gabriele Nardone, Vincenzo Piscopo, Enrico Zambianchi

**Affiliations:** 1Dipartimento di Ingegneria Meccanica, Energetica, Gestionale e dei Trasporti, University of Genova, Via Opera Pia 15A, 16145 Genova, Italy; 2Dipartimento di Scienze Biologiche, Geologiche e Ambientali, Università degli Studi di Catania, Corso Italia 57, 95129 Catania, Italy; andrea.cannata@unict.it; 3Istituto Nazionale di Geofisica e Vulcanologia, Sezione di Catania, Osservatorio Etneo, Piazza Roma 2, 95125 Catania, Italy; 4ARPAL-Hydrological and Weather Centre, Viale Brigate Partigiane 2, 16129 Genova, Italy; antonio.iengo@arpal.liguria.it; 5Department of Engineering and Department of Science and Technology, Università degli Studi di Napoli “Parthenope”, Centro Direzionale Isola C4, 80143 Naples, Italy; maurizio.migliaccio@uniparthenope.it (M.M.); vincenzo.piscopo@uniparthenope.it (V.P.); enrico.zambianchi@uniparthenope.it (E.Z.); 6Istituto Nazionale di Geofisica e Vulcanologia, Sezione ONT, Via di Vigna Murata 605, 00143 Roma, Italy; 7ISPRA—Istituto Superiore per la Protezione e la Ricerca Ambientale, Via Vitaliano Brancati 48, 00144 Rome, Italy; gabriele.nardone@isprambiente.it; 8Consorzio Nazionale Interuniversitario per le Scienze del Mare (CoNISMa), Piazzale Flaminio 9, 00196 Rome, Italy

**Keywords:** dynamic measurement, sea state measurement, wave buoys, satellite remote sensing, coastal radars, shipboard sea state observation, microseism observation, networks for sea waves monitoring

## Abstract

Sea waves constitute a natural phenomenon with a great impact on human activities, and their monitoring is essential for meteorology, coastal safety, navigation, and renewable energy from the sea. Therefore, the main measurement techniques for their monitoring are here reviewed, including buoys, satellite observation, coastal radars, shipboard observation, and microseism analysis. For each technique, the measurement principle is briefly recalled, the degree of development is outlined, and trends are prospected. The complementarity of such techniques is also highlighted, and the need for further integration in local and global networks is stressed.

## 1. Introduction

Sea waves are produced as a response to wind energy transfer at the air–sea interface. Short surface waves form at the sea surface, increasing the surface roughness and, thus, the wind stress and the wave height. This process continues until the waves have reached equilibrium with the wind forcing. In a vertical plane, sea waves are formed in two extremes: wave crests and troughs. The vertical distance between crest and trough is defined as the wave height, whereas the wave period can be defined as the time it takes for two consecutive crests to pass through a fixed point. Wave sizes vary from centimeters in length (ripples or capillary waves) to kilometers (storm surges and tides), but historically, the measurement of waves in the open sea has aimed at recording information on wind waves, with a wavelength from meters to hundreds of meters.

An apparently random sea surface can be thought of as the sum of many simple wave trains, whose parameters can be defined via a time domain approach (zero-crossing). However, introducing the wave spectrum in the frequency domain is a more efficient way to formalize this concept. Using harmonic analysis, each wave recording is broken down into a large number of sine waves of different frequencies, directions, amplitudes, and phases. This approach, defined as Fourier analysis, provides an approximation of the irregular shape of the real sea wave, recorded as the sum of trigonometric functions (sine curves); each frequency and direction describes a component of the wave that has an associated amplitude and phase.

The wave height is usually expressed as significant wave height *H_s_*, defined as the mean value of the highest one-third of wave heights [1], or it can be estimated from the spectrum obtained from a time series of sea surface elevation. The vertical displacement of the sea surface over time in a fixed position, measured with a non-directional instrument, can be represented as a sum of sinusoidal signals in frequency *f*. Assuming random phases and adding the square of all the amplitudes in a small frequency range, we obtain a non-directional wave frequency spectrum *S*(*f*) of the wave signal (with dimensions m^2^/Hz).

Wave spectra can be estimated using spectral estimation methods—either non parametrical, such as the fast Fourier transform (FFT) following a proper estimation procedure [2], or parametrical, based, for example, on autoregressive moving average (ARMA) modeling of the time history [3].

Directional instruments can also measure horizontal displacements. The goal of directional wave measurement is to obtain accurate estimates of the two-dimensional energy distribution in frequency f and direction θ, without any preliminary assumptions about the shape of the distribution. The sea surface can be described by the two-dimensional spectrum of waves in frequency and direction *S*(*f*,*θ*), expressed as the product of the non-directional wave frequency spectrum *S*(*f*) and the directional distribution, as follows:(1)$$S\left(f,\theta \right)=S\left(f\right)\left[a_{1}\cos\theta +b_{1}\sin\theta +a_{2}\cos\left(2\theta \right)+b_{2}\sin\left(2\theta \right)+\sum_{n=3}^{\infty }\left(a_{n}\cos\left(n\theta \right)+b_{n}\sin\left(n\theta \right)\right)\right],$$
where *n* is the summation index. It is currently highly recommended that all directional wave measurement devices reliably estimate the energy of the wave *S*(*f*), which is related to the wave height, and the first four coefficients of the Fourier series *a*_1_, *b*_1_, *a*_2_, and *b*_2_ in Equation (1), which defines the directional distribution of this energy [4]. The combination of *S*(*f*), *a*_1_, *b*_1_, *a*_2_, and *b*_2_, or any other equivalent parameters [5], forms the set of “first-5” spectral wave parameters. They provide basic information (significant wave height, peak wave period, and average wave direction in the peak wave period), as well as a further set of sea state information to be used for a wide range of applications. Furthermore, the first four moments of the directional distribution are the mean direction of the wave (the first moment), the directional spread (the second moment), the skewness (the third moment, defines how the directional distribution is concentrated), and the kurtosis (the fourth moment, defines the peakedness of the distribution).

Significant advances have been made in the measurement of waves over the past decades, and numerous measurement devices are now available that operate on different principles, some of which are well-established while others are still under development. Here, several measurement techniques are reviewed, including buoys, satellite observation, coastal radars, shipboard observation, and microseism analysis. For each technique, the measurement principle is recalled, the degree of development is outlined, and trends are prospected. In any case, standardized measures are essential to ensure consistency between the different stations, so much so that it is necessary to define reliable measurement networks and integrate information at the regional level. This last aspect is thus highlighted at the end of the paper

## 2. Wave Buoys

### 2.1. Drifting Buoys

Buoys, whether moored or drifting, have the ability to communicate, in a programmed way, in real time via satellite telecommunication systems, transmitting acquired observations to collection centers. The collected data are used in different applications, including research, environmental monitoring, weather forecasts, validation of oceanographic and meteorological models, as well as safety at sea and coastal defense works planning [6].

There are many different types of drifting buoys and moored buoys, depending on the application and measurements required in different seas and oceans. Drifting buoys are floating platforms without any type of anchor, and can be divided into three different categories: surface drifter, subsurface floats, and ice buoys. Surface drifters (see Figure 1) provide a unique representation of surface current dynamics, and can supplement satellite observations to study climate-scale problems. Generally, drifters are characterized by a surface buoy and a drogue moving below the sea surface and attached by a long, thin tether. Batteries, sensors, and other electronics are contained in the buoy, whereas the drogue is an underwater anchor, a cylinder of four to seven sections, with a large hole through the middle of each section, giving the drogue the appearance of a holey sock [7]. The holes are important because they create lots of small areas of turbulence to slow its larger slipstream speed and improve its stability when moving. In this way, its speed and direction can be made to better match those of the actual currents. Drifters also measure temperatures, salinity, air pressure, and surface wind speed and direction. A drifter that is not tethered to its drogue is a good wave following device, and is a potential tool for global wave measurements. A drifter can also be used to yield high-quality directional wave spectra by installing on it a downward-looking acoustic Doppler current profiler (ADCP), used to derive two-dimensional spectra from wave orbital velocities, or a global positioning system (GPS), to measure the motion of the drifter at frequencies greater than 0.01 Hz. Whereas ADCPs are expensive, GPS sensors are relatively cost-effective and easy to install. Thus, wave drifters are generally developed based on GPS receivers and deployed in the open sea. Directional wave spectra (DWS) drifters are a new generation of GPS-based tracking devices, able to compute the “first-5” directional Fourier coefficients (a_0_, a_1_, b_1_, a_2_, b_2_) used to derive wave parameters such as significant wave height, swell direction, and directional spread, among others [8].

Autonomous systems used to profile the deeper waters of oceans are called floats, similar to drifters, because they are unmoored and measure currents, temperature, and salinity. Floats are programmed to drift below the sea surface at different depths, forced by both horizontal currents and surface waves. Measuring waves accurately using floats is possible, and relatively inexpensive. One of the main applications involves an Inertial Motion Unit (IMU)-type device consisting of tri-axial accelerometers, rate gyros, and magnetometers. IMUs provide an accurate measurement of accelerations and tilts. Further upward-looking ADCP is also applicable. To change their buoyancy, floats are equipped with simple mechanical pumps, bladders, and other devices, allowing them to fluctuate between different depths. Data sending takes place after the floats rise to the sea surface periodically to send data via their satellite antenna [9].

The main application of both surface drifters and subsurface floats is to measure ocean trajectories, which are useful for both visualizing ocean motion and determining the time-evolving velocity fields. Lagrangian analysis of ocean velocity data reveals that very different kinds of circulation patterns occur in different regions, and shows the interactions between currents, topography, and coastlines. A recent experiment has shown the possibility of obtaining wave measurements from floats via measurement of the pressure difference between the top and the bottom of the float [10].

Ice buoys are mainly used to track the dynamic and thermodynamic evolution of drifting Arctic and Antarctic sea ice, and to acquire meteorological and upper oceanographic data [11,12]. They can sit on sea ice in the open Artic or Antarctic Ocean to evaluate the seasonal evolution of the thermohaline structure of the ocean during ice formation/freeze up. Depending on how the platform sits on the sea ice, snow depth and ice thickness variations may be derived from either ultrasonic range finders, such as those on automatic weather stations (AWS) or ice mass-balance thermistor strings, or solid-state sensors [13,14]. Generally, ice buoys measure the ice motion using an inertial motion unit (IMU), performing measurements at 10 Hz and transmitting the full wave spectrum, geographical location, and battery power status at predefined intervals [15]. In most cases, these buoys are part of a wider buoy network, which includes moored wave buoys, automatic weather stations, GPS buoys, and data loggers that transmit the data via satellites in real-time. In research applications, ice buoys used for wave spectral data detection via onboard accelerometers have been deployed in marginal ice zones during the period of ice formation (pancake ice) [16].

### 2.2. Moored Buoys

The last type of buoys focused on in this paper is moored buoys (see Figure 2), a category that encompasses a large number of platforms (either small and cheap or relatively large and expensive) anchored in fixed positions to derive long-term observations under many various atmospheric conditions and with different oceanographic sensors.

Several platforms have been developed for moored buoys; they can vary from a few centimeters in height and width to over ten meters, but all different sorts of moored buoys had been developed to capture and model information about ocean dynamics on the surface, determining the directional spectrum of waves in the open sea. Data are usually transmitted in real time and disseminated via the global telecommunication system (GTS) of WMO for use by national meteorological centers.

These data can be used to greatly improve forecasting and warnings for severe storms, since wave patterns have been verified to exhibit negative bias at maximum wind speeds [17,18,19].

A buoy wave-measurement system has some typical components, such as the platform (comprising the hull and the mast), the power system (e.g., sealed lead acid gel batteries charged by solar panels), the electronics payload (i.e., data acquisition systems, nautical light, GPS receivers, and one or more data transmitters), the sensors (wave and meteo-oceanographic sensors), and the mooring. Wave buoys can be spherical, cylindrical, discus-shaped, or boat-shaped [20]. Plastic and foam construction has spread in recent years for hulls, but the current trend is to use aluminum or marine alloy steel as a construction material to prevent the release of microplastics into the sea. Mooring methods depend on the depth of deployed waters and cost factors; essentially, there are three typical systems: all chain mooring (used in shallow water), semi-taut mooring, and inverse-catenary mooring [21,22]. The buoys must be moored carefully; tensions are to be avoided so that their movements are not affected by the mooring and they can adequately define the free surface. Moorings that allow the buoy to float freely and actively rotate within a well-defined guard circle can be made of steel-linked chain and wire rope, synthetic fiber rope, or bungee cord. Synthetic ropes (nylon, polyester, polypropylene, or advanced fibers) do not noticeably corrode or deteriorate in seawater, and their strength to immersed weight ratio is excellent, so they are often used for buoy moorings.

Several methods have been described to determine the directional spectrum of waves in the open sea using moored buoys; a full account of the physical principles on which they are based has been given in [23,24,25,26], while an in-depth study of the evolution of these devices was given in [27,28,29]. The spectrum of the sea surface is not exactly reproducible analytically, but under certain wind conditions the spectrum acquires a characteristic shape. With experimental tests, parametric spectrum models have been obtained consisting of approximate expressions that can adapt to the spectrum of sea surface elevation. Among the many proposed similar models are the Pierson–Moskowitz model and JONSWAP [30]. These spectra were experimentally derived under fully developed wind conditions in the generation area in deep water, and can be used to relate the shape of the spectrum of the wind waves. To measure the directional spectrum of wind waves using moored buoys, an FFT is generally performed on the buoy displacement data. Considering only the “first-5” Fourier coefficients, this produces a smoothed version of the directional spectrum, as the coefficients are averaged to decrease the variance of the estimate. After that, the directional wave spectrum is calculated from the average coefficients using an appropriate weighting function [31,32].

Moored buoys either follow water particles (typically spherical shaped), referred to as wave-following or translational buoys, or they follow slopes (typically disc-shaped), referred to as surface-following or pitch and roll buoys. These essentially involve a set of inertial sensors, such as accelerometers and tilt sensors, for the determination of heave pitch and roll angles [28]. Accelerometers are used to detect the axial acceleration of the buoy concerning the inertial system, which is then used to calculate the real-time position of the buoy by integrating twice. The gyroscopes acquire the reference coordinates that are essential for the aforementioned positioning. Gyroscopes require no external information and have no optical or electrical connections to the external environment. However, because gyroscopes contain weak parts that rotate at high speeds and can produce drift, they are less reliable for long-term high-precision operation.

These sensors’ wave height and directional frequency spectra estimations are based on measurements of three concurrent time series, which can be transformed into a description of the sea surface via a 6-degrees of freedom non-linear equation of motion in the sensor to provide good integral wave parameter estimates (height, peak period, mean direction at the peak period, etc.), and by using the Maximum Entropy Method, an adaptive data procedure capable of obtaining a spectral estimate of higher resolution than FFT based on shorter data records, which fits an autoregressive model to the data to derive directional frequency spectra [33]. A triaxial accelerometer measures acceleration in three mutually perpendicular directions. It does so using a single accelerometer unit measuring the acceleration along each axial direction (*x*,*y*,*z*), while double integration of the triaxial acceleration derives the buoy motion along (*x*,*y*,*z*). Sensor systems can sample at rates ranging from approximately 1 Hz to 10 Hz or more; furthermore, the sampling period can vary from about 17 min to more than 35 min. These variations all contribute to the differences in the measured waves.

In the most widely used water particle-following buoys, a heave–pitch–roll sensor is mounted, along with two horizontal hull-fixed accelerometers and a compass to determine directional wave information. A common alternative configuration uses three accelerometers to measure total accelerations along the mutually orthogonal axes of the buoy (*x*,*y*,*z*), and three angular rate sensors to measure rotation rates; slope-following buoys use a similar sensor package. Pitch and roll angles, *dz*/*dy* and *dz*/*dx*, are crucial for deriving directional wave information from data buoys; a gimbaled gyroscope sensor is typically used to provide the pitch and roll information, but this may also be derived from the outputs of a triaxial angular rate sensor, using magnetometers or a dedicated motion package [34]. The Fourier coefficient estimates are derived directly from the measured accelerations and linear wave theory [35,36].

For a slope-following system, the coefficient estimates must include various corrections for the mooring response of the hull, which is trigonometrically related to the four Fourier directional coefficients [37,38]. To calculate the directional wave parameters, the buoy will use both angular motion data and surface elevation data in the frequency domain by measuring the correlation between the slopes of the north–south and east–west angles and the lift dataset. Then, it will calculate the directional coefficients. The vertical direction used is that along the axis of the buoy itself, rather than the true vertical in relation to the Earth, but this can cause serious errors. The gimballed sensors have an integrated mechanical system used to keep the accelerometer vertical when the buoy and the sensor are tilting. These have been used operationally since the early 1970s to determine pitch and roll angles [39].

In recent years, the trend has been to migrate from these fixed and gimballed accelerometers to electronic motion packages housing triaxial accelerometers combined with digital magnetometers and compass packages that cover nine degrees of freedom, in order to measure the motion of the buoy and translate it to the free surface. The cost, power consumption, and size of these are significantly lower, and the greater challenge remains the returning of high-quality Fourier coefficients for directional estimators. Strap-down accelerometers, due to their low cost and low maintenance requirements, are a very common type of sensor. Compared to a traditional gimballed accelerometer, a strap-down accelerometer significantly reduces the cost and size of the system and does not require any external inputs or devices. However, due to errors in manufacturing and propagation errors in the onboard calculations, a strap-down accelerometer can produce a significant error, especially in low-cost systems [40]. These mechanical and electrical components can be combined, on a micrometric scale, to obtain a microelectromechanical system (MEMS). The analysis package (payload) acquires the raw signal and transforms it into an estimate of the free surface, from which directional estimators (the Fourier coefficients *a*_1_, *b*_1_, *a*_2_, and *b*_2_ in Equation (1)), frequency spectra, and integral wave parameters are ultimately derived.

To derive more accurate measurements, it would be preferable to have buoys built to carry out wave measurements only, because in this way the hydrodynamic response of the hull and the software filters used can be optimized. However, for reasons of cost-effectiveness, many buoys are designed to measure more parameters than wave data, especially in deep water, so the accuracy of these buoys’ wave estimates can be significantly compromised (for directional and non-directional waves). In fact, since the information on directional waves is derived from the movements of the buoy via the motion sensors installed onboard, through a mathematical transfer function that resolves the response of the buoy to the wave motion, it is quite evident that some specific characteristics of the buoy, such as the shape and dimensions of the hull and the superstructure, the material composition and the characteristics of the mooring, have a considerable influence over the estimate of the free surface. This is especially noticeable at low energy levels and when the measured wave signal is weak due to increased signal noise, such as in the cases of short and long wave periods. As has been pointed out, although it is known that each buoy should have its transfer function calculated for the deployment configuration based on the aforementioned physical factors affecting its movement, in most cases, this is not feasible [41]. Many buoys are developed as a part of projects for which a single transfer function is generally implemented according to the design conditions. It therefore becomes important to determine the transfer functions necessary to correct the variation caused by waves when the moored buoys do not exactly follow the surface of the wave. For example, it is necessary to compensate for the response of large buoys [40]. A comparison test of six different buoys with a variety of manufacturers, shapes, sizes, sensors characteristics, and mooring configurations showed general agreement for the integral wave parameters, but significant differences for the spectral parameters [42].

A recent development in buoy sensors is to derive wave parameters and the directional wave spectrum using the buoy GPS output velocities, representing the velocities of water particles. Comparisons of simultaneous GPS-derived and buoy-measured directional wave spectra showed good agreement [43]. There are numerous studies comparing the advantages and disadvantages of buoys using GPS and traditional buoys using accelerometers, highlighting that the levels of accuracy and the fields of application of the two technologies are very close [44,45].

Using a GPS receiver to measure wave parameters does not require subsequent calibrations or complicated processing to extract the wave data (it can be done by applying a high pass filter). GPS buoys, which are typically more compact in size, can facilitate three-dimensional movement with an accuracy of 1–2 cm up to wave periods of 100 s, and an accuracy of between 0.5% and 1% of the measured value. On the other hand, the energy consumption of the buoy is high, and moreover, errors and distortions of the GPS measurements are possible due to the multipath of the antenna and interference from various sources. Wave measurements using GPS have been pioneered by several research groups and manufacturers [46], but recent advances in technology and lower prices are paving the way for global implementation [8].

## 3. Satellite Remote Sensing

In this section, the physical basis and logic components of satellite microwave remote sensing are reviewed first. The goal is to provide a unitary and solid framework of the fundamentals. In Section 3.2, a brief summary of the main value-added products, out of the ones focused on the sea wave, is given. Special reference to operational services is made. Finally, in Section 3.3, the use of microwave remotely sensed measurements in sea waves monitoring is reviewed.

### 3.1. Background

Satellite microwave (MW) remote sensing represents a special observational tool especially for regions that are not easily accessible for in-situ measurements. Further, due to the MW’s propagation and interaction peculiarities, satellite measurements are independent of solar illumination providing a denser revisit time [47]. This is of particular relevance to the ocean environment, since it is the main locus of dynamical processes, and in some areas only a few in situ measurements can be made, for instance in the Southern Ocean. Furthermore, MWs are much less dependent on cloud cover [47].

MW remote sensing also involves some critical features that must be considered. Satellite MW remote sensors are narrowband systems that estimate the complex reflectivity of the ocean surface in an “imperfect” manner. The main objective criteria for defining such imperfections are the resolutions—the radiometric resolution, the spatial resolution and the spectral resolution. The additional temporal resolution or revisit time depends on the sensor swath and scanning configuration, as well as the platform height.

MW satellite sensors can be passive or active: in the passive case, when there is no source of illumination onboard the satellite, the electromagnetic data received at the sensor antenna are those naturally emitted by the marine scene; in the active case, an electromagnetic pulse is transmitted by the sensor antenna into the ocean, and the corresponding electromagnetic signal reflected by the environment is received by the sensor [47]. In both cases, random measurements occur because of the random nature of the scene.

Another important point is the observational scale. Some satellite sensors are meant to observe geophysical phenomena at large scales, and therefore they are called large-scale sensors, while some others operate at small scales, and are called small-scale sensors. In general terms, the spatial coverage is paid for by a coarser spatial resolution [47].

MW remote sensing calls for the solution of three subproblems: measurements, forward modeling and inverse modeling. The first subproblem is relevant to the accomplishing of not only high-quality measurements, but also to the proper design of the sensor. The second subproblem is sometimes neglected in modern blind approaches; once a proper geophysical quantity of interest is defined, it is important to investigate whether the observed measurements are related to said geophysical quantity. In a formal sense the forward model calls for the development of a theoretical relationship between the observable quantity at the sensor and the geophysical quantity. Further refinement leads to a semi-empirical geophysical model function (GMF) that is appropriately tailored with reference to the geophysical quantity and the sensor of interest. The third subproblem is known as the inverse problem, and calls for the quantitative estimation of the geophysical quantity of interest via a proper set of measurements. This latter problem is usually non-linear, ill-posed, and affected by noise. Despite these challenges, MW remote sensing is able to generate high-quality geophysical products that greatly contribute to the advancement of marine science and operational services.

Passive MW sensors are known as MW radiometers, and collect the electromagnetic field data naturally emitted by the environment at their receiving antenna. MW radiometers are large-scale sensors, and their spatial resolution is not fine [47].

Radar altimeters are large-scale nadir-facing active sensors. This peculiar nadir-facing configuration and its spatial resolution makes the measurements sensitive to large-scale sea surface slopes. Therefore, the radar altimeter is the best MW sensor for sea state estimation [34,48,49,50,51].

Scatterometers are large-scale off-nadir active sensors. The primary application of scatterometers is the indirect measurement of near-surface (local) winds over the ocean. Such scatterometer winds are routinely assimilated into Numerical Weather Models [52,53,54,55].

Synthetic Aperture Radar (SAR) is a narrowband-coherent, i.e., phase-preserving, off-nadir-active MW imaging sensor. The design of SAR is meant to enhance the spatial resolution, and it is used for applications that are not time-variable within the coherence time or integration time of such sensors. The SAR is a small-scale sensor that, although it is mainly optimized for defense and geophysical applications, has gained more and more interest in marine applications as well [47].

### 3.2. Review of the Main Marine Value-Added Products

Let us briefly review the main marine value-added products associated with the MW satellite sensors but the ones related to sea waves that are reported in much more detail in Section 3.3.

Standard multichannel MW radiometers have different beams, each associated with a frequency, and for each beam the horizontal/vertical polarization powers are measured. The combination of the different channels, each characterized by a frequency and a polarization, allows for producing some key operational added value products for marine applications. Satellite MW radiometers provide a valuable picture of the global sea surface temperature (SST) at about 25 km [56]. In other cases, SST maps are also obtained by MW and infrared radiometer measurements. At the same scale, MW radiometers operationally provide wind speed maps [57]. The accuracy of such latter products has been assessed in various ways, and in rain-free cases, matches that of an active sensor [58,59]. MW radiometers are routinely used to monitor hurricanes [60].

Two special classes of advanced MW radiometers have also been developed. The first class includes the fully polarimetric radiometer onboard the WindSAT, which is used to estimate the sea surface wind field [53,61]. In the second class we find the L-band MW radiometers, i.e., those onboard the European Space Agency (ESA) SMOS and NASA Aquarius satellites [62,63]. They are designed to accurately estimate the sea surface salinity at a global scale.

The gold standard for estimating the sea surface wind field is provided by the scatterometer. The estimation of the wind field from such measurements requires significant processing, and involves the solution of a nonlinear inversion problem. The mathematical core of the first phase calls for the minimization of an objective function—that is, a convex function of the residual between the measurements and the GMF [54]. The wind estimation approach results in multiple solutions associated with local minima in an objective function, formed from the noisy backscatter measurements. The second step is known as the dealiasing step, and allows the selection of the best solution. It has been demonstrated that in the set of possible solutions, or aliases, there is a true solution, and that it is possible to identify it with only external information [54]. Scatterometer winds are routinely assimilated into numerical weather models [59].

The Synthetic Aperture Radar (SAR) is a small-scale off-nadir active imaging sensor that integrates a set of echoes to achieve a fine spatial resolution. In very simple terms, such integration can be described as an offline processing approach that coherently combines the echoes properly corrected by the different traveling paths [47]. The calibrated complex image is the product used most commonly to extract the geophysical information of interest. In the case of the SAR, the electromagnetic modeling of sea surface scattering calls for a long wave and short wave sea surface model. Furthermore, in the case of SAR, ocean dynamics affect the image formation, and different models for this have been proposed in the literature [64]; the main dispute is between the distributed surface (DS) [65] and the velocity bunching (VB) model [66]. Because of the dynamic influence of the ocean environment in SAR imaging, a debate on focus adjustments emerged in the early days [67], but nowadays, SAR image processing is performed for a static scene, and so the residual information is retained [67].

The marine applications of SAR measurements are diverse, but the only one that has been considered to support operational services is related to marine oil pollution and vessel monitoring. Such applications are particularly benefitted by polarimetric SAR measurements [68,69].

### 3.3. Sea Waves Monitoring

In this subsection, the focus is on ocean wave products as generated by MW satellite remote sensing measurements.

The gold standard for the estimation of the Significant Wave Height (SWH) is the radar altimeter. It transmits pulses into the sea surface to accurately estimate the distance between the satellite and the sea. The electromagnetic interaction in this case is governed by the Kirchhoff scattering model, showing a dependence on large-scale sea slopes.

Two main physical problems must be considered. First, the interpretation of the (averaged) pulse time delay in terms of distance, then the semi-empirical modeling of the received echo and its use in the inversion process.

Accurate range estimation calls for two main subproblems to be considered: the precise orbit determination of the satellite, and the effective electromagnetic wave propagation speed. The first subproblem requires the use of precise positioning measurements. This benefits from high-precision positioning signals, such as GPS [48]. Furthermore, the use of high satellite altitudes allows us to limit the atmospheric drag, to obtain a more regular Earth gravitational field and to better track the satellite from the ground.

The accurate estimation of the range depends on the corrections of the vacuum speed [48]. The main atmospheric corrections are required due to the ionosphere and troposphere, the latter of which is characterized by two terms: dry tropospheric correction and wet correction. Other corrections are related to sea surface scattering, and are collectively known as sea state bias. In order to most effectively estimate such corrections, dual-frequency radar altimeters are typically used, and a side radiometer sensor is deployed onboard [48].

With reference to the semi-empirical modeling of the received echo, it must be underlined that the radar altimeter is optimized for the open sea, and the reference average echo waveform is the Brown model [48,49]. The echo waveform can be used to estimate the free parameters of the model by comparison with real (averaged) measurements. Such an inversion process is known as retracking, and allows us to estimate the range with centimeter-level precision, as well as the SWH [34,48,49].

The fluctuations in range measurements due to tides, atmospheric pressure, and ocean waves must be filtered out to estimate the sea surface height (SSH). The deviation in SSH from its mean over a few decades is known as the sea level anomaly (SLA).

In Figure 3, the global SWH map measured by the ESA ERS-2 radar altimeter during the 1995 boreal summary is shown.

Currently, efforts are being made to enhance the radar altimeter inversion process in coastal regions [49,50,51]. There are two main approaches that may co-occur in real life; the first seeks the resolution enhancement of the measurements by exploiting the partial correlations of the return, and the second looks for special forms of the semi-empirical waveform model [49].

The radar altimeter-derived SLA and SWH products are assimilated at ECMWF. These products are also used for climate studies; see Figure 4.

We now consider the use of SAR measurements. It is here necessary to underline some special imaging characteristics of the SAR. It is a phase-preserving, i.e., coherent, imaging sensor, which, given its proper SAR processing chain, generates fine-spatial resolution images by means of two very different scanning mechanisms: in the range or across-track direction, the spatial resolution is given by linearly modulated chirp pulses traveling back and forth at the speed of light, i.e., effectively instantaneously; in the azimuth or along-track direction, the spatial resolution is given by the platform motion that composes the temporal long array, i.e., at a velocity that is appropriate for time-varying environments [70].

Since the SAR is a coherent sensor, within each resolution cell there occurs a physical phenomenon known as fading, which can be modeled as the sum of independent elementary scattering centers that distinguish each other within sea-free scenes via the scene’s micro roughness, i.e., at the scale of the electromagnetic wavelength. The manifestation of such a fading process on the SAR image is known as a speckle [70]. Although the speckle is often taken to be uninformative, and in several automatic procedures it is reduced at expense of spatial resolution, this is untrue, and in ocean scenes, it can be related to the sea state [71,72].

Because of the SAR’s incident angles, the small-scale backscattering is, for low to moderate sea states, modeled by resonant Bragg backscattering, i.e., due to ripples in the range of the microwave wavelength. Hence, longer waves are imaged indirectly under amplitude and phase modulation processes, known as Real Aperture Radar (RAR) and motion-induced effects, respectively [73]. The RAR process can be described by a linear function (weak modulation), which relates the NRCS to the long sea wave field: the RAR Modulation Transfer Function (MTF) [73]. The RAR MTF is modeled by three terms: the tilt modulation term, the range bunching modulation term and the hydrodynamic modulation term. The motion-induced effects are SAR-specific mechanisms, and are due to the SAR azimuth channel acquisition mechanism. The radial component of the orbital motion associated to the long sea waves generates an extra Doppler shift with respect to stationary scenes [73], giving rise to velocity-bunching phenomena. In fact, since the scattering elements are characterized by different orbital velocities, they are non-uniformly displaced in the SAR image plane, and so the apparent positions of the scattering elements are bunched and spread out. The radial component of the orbital acceleration is responsible for the degradation of the azimuthal resolution. Since both orbital acceleration and orbital velocity vary along the flight direction, they can produce a wave-like pattern on SAR images. However, for certain radar and sea parameters, the wave pattern can be severely distorted or completely smeared out [73,74].

For azimuthal traveling waves, the imaging may be highly nonlinear, while the imaging process is always linear for range-traveling waves and for quasi-range-traveling waves [73,74].

Hence, the estimate of the sea directional spectrum via SAR images is not a trivial task. The leading paper on the subject is [75], wherein an iterative ill-posed procedure is described. The inversion algorithm was refined in [76].

Important advancements have been made in this area, and are described in [77,78,79,80]. In [77,78], a new approach exploiting the cross-spectra of the individual SAR is presented and discussed. The real part of the SAR cross-spectra is exploited to retrieve the ocean wave spectra, while the imaginary part is exploited to solve the SAR-inherent 180° ambiguity. In [79], further advancements on the cross-spectra approach have been presented. The analysis shows that the benefits of Sentinel-1 SAR high-quality wave-mode measurements can be further extended towards shorter-scale waves.

Although classical techniques provide reasonably accurate wave measurements, especially for swell waves, they face two main problems: the rough knowledge of the RAR MTF, and the need for reliable a priori information to make the inversion process convergent.

A new parameter, called MACS, is also defined, and is associated with the range-detected ocean wavelengths of 15–20 m [79]. This parameter has the advantage of not calling the non-linear inversion scheme [75,77]. In [80], this parameter is exploited for global analysis. Along the same conceptual idea has been designed one the sensor onboard of the China France Oceanography Satellite (CFOSAT): the SWIM. It is a Real Aperture Radar (RAR) that observe range travelling waves at high spatial resolution while filter out the azimuth travelling waves [81].

In [82], a study based on deep-learning and co-located radar altimeter and ESA Sentinel-1 SAR data offers remarkable results. The deep learning approach has been implemented for low-level SAR cross-spectra, used to estimate the SWH.

In [83,84,85], fully polarimetric SAR measurements are exploited. Physically, the approach benefits from the dependence of the polarimetric Cloude–Pottier decomposition, i.e., the eigenvector dependence, on the orientation angles. Such an approach is feasible for high-quality fully polarimetric SAR sensors, such as Radarsat-2. The main advantage of this approach is that the complex hydrodynamic MTF [74] does not need to be estimated [83,84]. In [85], a validation of the polarimetric approach with Radarsat-2 measurements is performed. The analysis shows good agreement with buoy data [85].

Because of the non-linear relationship between the SAR image spectra and the ocean spectra [73,74,75], several empirical approaches have been explored to estimate the SWH using SAR images, e.g., in [86,87], two polarimetric approaches are presented.

A popular and effective approach is the so-called azimuth cut-off approach [88,89,90,91,92,93]. It was first proposed in [88]. It uses physical phenomena directly associated with the SAR azimuth channel’s image formation [73,74,75]. In very simple terms, the azimuth SAR channel is unable to image ocean wavelengths smaller than the azimuth cut-off. Such a cut-off is empirically related to the sea state, and in some cases it can also be interpreted in terms of wind speed [89,90]. The actual implementation is rather complex, and affects the final quality of the estimate [90], but the enhanced quality of the new SAR sensors, such as that onboard the ESA Sentinel-1 missions, makes the azimuth cut-off approach very promising.

Let us finally consider another method for observing sea waves: along-track SAR interferometry [94,95,96,97]. The along-track SAR interferometer measures complex image correlation using two SAR acquisitions that are in all respects similar except for the (small) time difference. The phase difference, i.e., the along-track polarimetric phase, can be related to the ocean wave spectra, allowing its estimation. In fact, the sea surface radial velocity can be determined by the interferometric phase of each resolution unit, and then the wave spectrum is obtained from the radial velocity. However, this method is still affected by velocity bunching.

An alternative approach, based on across-track airborne SAR interferometry, has been also proposed [98]. It employs a well assessed procedure meant to estimate the Digital Elevation Model over stable scenarios to the marine case. Of course it must be operated with the two SAR antennas of the interferometer acquiring at the same time (single-pass mode) and this is not feasible by satellites at present time [98].

The fundamental advantage of these methodologies is related to the fact that while the amplitude spectra depend strongly on the NRCS modulation of ocean long waves, which is roughly known, the phase difference is related to the ocean wave spectrum in a known manner.

Since SAR has the unique ability to indirectly measure the sea’s directional spectra, it is important to assess the quality of such SAR-derived wave spectra [99]. In [99], a quality analysis of the key characteristics is performed based on a physical/statistical approach.

## 4. Coastal HF Radars

Coastal HF (High-frequency) radars are land-based remote sensing instruments that have attained great popularity in the last few decades, even though they are still considered an “emerging” observation technique by the Global Ocean Observing System. The reason for their wide distribution [100] lies in the fact that they are able to provide synoptic data (i.e., simultaneous over a relatively large area), repeated in time at unprecedentedly high spatial and temporal resolutions.

Their essential principle is based on the backscattering of an electromagnetic signal by the sea surface. This phenomenon gives rise to a spectrum of backscattered signals (see Figure 5). The main peaks correspond to the first-order scatter from the so-called Bragg waves. They are the result of a coherent resonance, analogous to the Bragg effect that emerges in atomic lattice detection by X-rays. These Bragg peaks occur when the wavelength of surface ocean waves is approximately half as long as the wavelength of the transmitted signal, as first observed in [101] and later clarified in [102,103,104,105,106]. These first-order peaks provide information about surface currents, whose velocity (radial velocity with respect to the antennas) can be inferred in relatively simple terms from the Doppler shift associated with the presence of a current underlying the surface wave trains, as reviewed in [107].

The continuum surrounding the first-order (dominant) peak represents the higher-order scattering, partly due to the nonlinear interactions between surface ocean waves. This portion of the spectrum is the source of information regarding surface gravity waves. Its inversion is based on the relationship, established in [108,109,110] and further developed in [111,112], between the backscattered signal and the ocean wave directional spectrum. The inversion of an integral equation allows us to retrieve wave parameters such as wave height and mean direction, as well as dominant wave period. Several approaches have been applied to carry out such a reconstruction, as reviewed in [113].

The inversion process is far more complex than that of the first-order spectrum, and is subject to a number of theoretical limitations, as thoroughly discussed in [115].

At the lower end of the measurable wave height range, i.e., at low sea states, the main limitation is related to the low signal-to-noise ratio, which is frequency-dependent (HF radars operating at higher frequencies can detect lower sea states), and this prevents the accurate detection of very low wave heights [115]. At the higher end of the measurable wave height range (high sea states), as well as in the case of very intense surface currents, first- and second-order peak regions may not be well separated, meaning the wave spectrum is not well defined. This effect is also frequency-dependent, and as a rule of thumb, the upper threshold for accurate wave height detection can be estimated by Equation (2):(2)$$h_{\mathrm{thr}}=\frac{2}{k_{0}},$$
where $k_{0}$ is the radar wavenumber [116,117]. Wave heights above $h_{\mathrm{thr}}$ will be under- or overestimated.

HF radars can be divided into two different categories, Direction Finding and Beam Forming systems [118]. Direction Finding instruments are characterized by a compact structure, with closely spaced or even co-located transceiving antennas; these systems provide information on the wave field that is not resolved in azimuth. This means that the outcome of the inversion in this case will represent an azimuthal average around the transceiving system, thus necessitating the use of circular statistics [119]. Parameters are thus derived over a circumference centered on the transceiving antenna’s location (the so-called range cell). Its radius should not be too long, to ensure a sufficiently strong signal, but not too short, to avoid breakers and the influence of local bathymetry [120,121]. On the other hand, Beam Forming, also known as Phase Array, radars utilize arrays of antennas, which introduces additional logistical issues in the installation and maintenance processes, and they are unable to reconstruct wave directional spectra on a grid.

As with any measurement instrument, and in particular remote sensing ones (even land-based, such as HF radars), validation is a non-negotiable prerequisite for scientific utilization. Validation proceeds through the intercomparison of measurements provided by different instruments. As discussed in [122], this is a very delicate issue, as metrics have to be devised to compare the outputs of different instruments that typically provide very varied results. A preliminary assessment of differences in the measurement principles, of their inherent constraints, and of biases due to different sampling rates and similar issues has to be preliminarily carried out. After considering the above, and comparable outputs have been produced, proper validation can be undertaken.

The typical touchstone of HF radar-derived wave parameters is represented by data obtained with different kinds of wave buoys [120,123,124,125,126,127]. However, other systems have been used for validation, including a bottom-installed current meter equipped with a pressure transducer [128] and satellite altimeter data [129]. More recently, the output of a sensor for directional wave measurements installed on an ADCP mounted on a MEDA elastic beacon was used [130]. Finally, impressive intercomparison experiments, utilizing several moored buoys along with a number of coastal weather stations and model outcomes, have recently been described [131].

The most common first validation step involves the superposition and qualitative comparison of data in time: this has been done in the past for relatively limited short periods, starting from the 1980s and early 1990s [132,133] up to very recently [134,135,136,137], with only a few exceptions, such as [138]. In recent years, such simple validations have been carried out in the framework of investigations over longer time periods, such as in the yearly analysis by [120], and in multi-annual ones [127,139,140].

The next (quantitative) step is building scatterplots and estimating statistical parameters, such as correlation coefficients [141]. This is also a very straightforward type of analysis for any data validation, and is extremely common, even though some caveats have to be taken into account [142]. Examples of simple correlation statistics of HF radar-derived wave parameters vs. in situ and/or remotely sensed data can be found throughout the relevant literature. A number of additional statistical descriptors used to validate HF radar data vs. buoy data (and vs. model outputs, see below) have recently been introduced [127].

A further step in the data validation process would be the comparison of frequency and directional spectra [115,143,144], which is not straightforward. As to the former, Krogstad et al. [122] underlined how the simple superposition of spectra measured by different instruments (e.g., buoy and HF radar) may not yield consistent results because of sampling differences, possibly solved by looking at the mean spectral ratio for specific frequency ranges and building some kind of spectral calibration on this basis. The direct comparison of directional spectra is also unlikely to provide robust results, but may nonetheless provide interesting information.

Once validated, HF radar data may, in turn, be used for the validation of numerical wave models, becoming the benchmark themselves, and thus inverting the perspective. This is the case reported in [145], which represents probably the first example of this, and such an approach was recently employed in the Gulf of Naples [130]. The latter paper compares HF radar-derived wave parameters with two different wave models, a coarse- and a high-resolution one. The authors find quite a good agreement between the two data sets, even though caution needs to be applied in this, as the above-mentioned theoretical limitations of the inversion procedures for radar data might affect the two extremes of observed sea states. By following this path and strengthening the sea-truth function, as was already done for surface currents [146], HF-derived wave parameters can also be assimilated into models [147,148,149].

To give a few examples of some recent results and applications, Figure 6 shows the results of a multiyear analysis carried out in the Gulf of Naples [127], where wave parameters were drawn from a network of three HF radar systems located in Portici, Castellammare and Sorrento (PORT, CAST and SORR in the map), utilizing data collected between May 2008 and December 2012 from a 5 km-radius range cell (red arcs in the map). Validation data for the seasonal variability, spanning from November 2015 through to December 2018, were derived from a sensor for directional wave measurements installed on an Acoustic Doppler Current Profiler, itself mounted on a MEDA elastic beacon just off the urban littoral of the city of Naples. The figure shows the modulation, in terms of direction and wave height, of the wave climate in the locations sampled, which show site-specific differences due to the bathymetry and morphology of the Gulf. A strong seasonal variability in the parameters shows up quite clearly, with higher wave heights in autumn and winter, as can be expected on the basis of the local and regional meteorological conditions (as detailed and discussed in the paper).

Figure 7 is an example of the use of HF radars during extreme events on the Spanish coast [121]: it shows very good qualitative agreement between buoy and HF radar-derived wave heights (SWH in the figure) in the course of two storms that occurred in 2017 and 2020 (Figure 7). The grey shaded columns and the black line are the hourly sea surface height (SSH) and the meteorological tide recorded by a tide-gauge located in Tarragona (TG1), respectively; the blue line represents wave buoy measurements provided by a Seawatch instrument deployed off Tarragona (B1); the red and green lines are the radar data derived with different versions of the proprietary manufacturer’s software used for the inversion (green line not present in the 2017 data). The pink dashed lines are the lower and upper limits of wave height detection derived from the theoretical constraints discussed above.

Figure 8 shows an application in wave energy extraction [140]; studies on the same line have been carried out in the past, in particular in and around the Wave Hub site, specifically developed for this purpose [145]. In such cases, the possibility of using the spatial distribution of directional spectra over a grid, rather than an azimuthal average, may be important, leading to the utilizing of phased-array HF radar systems. Figure 8 shows the monthly mean fields of wave potential distribution in an area off the coast of Chile, underlining the strong temporal and spatial variability. This variability, quantified in terms of indexes and coefficients in the paper, is an important piece of information for planning the installation of marine energy infrastructures.

As demonstrated by the above and several other papers over the last few years, HF radars have been proven to be able to provide reliable measurements of significant wave height, period and direction over shorter and longer periods, allowing us to highlight spatial and temporal variability in the wave climate, as well as details on the propagation of waves inside enclosed coastal areas, contributing significant information related to extreme storm events [121,124,127] and supplying valuable inputs for applications of various types. We have now realized that the multifaceted dynamics displayed by the coastal ocean call for integrated observation strategies. The present and the future of ocean measurement both lie in observation networks that complement, and interact with, each other. As illustrated in [100], in 2012, a global HF radar network was established and has been growing exponentially ever since. Even though it was planned primarily for monitoring surface currents, such a joint effort might effectively contribute to broadening the possibilities of measuring waves in synergy with other networks, helping us to grasp the richness of the phenomena developing in this extraordinary environment.

## 5. Shipboard Sea State Estimation

### 5.1. Brief Review of the State of the Art

The seminal idea of detecting sea state parameters using the onboard measurement and analysis of ship motions dates back to the mid-1970s, when the first pioneering research was undertaken by Takekuma and Takahashi [150] for ships without forward speed. The first attempts at including the ship speed and, consequently, the Doppler shift effect were made in the mid-1980s for ships in head and bow seas [151,152,153,154,155], and slightly later for ships advancing in following seas [155], including the well-known 1-to-3 multivalued problem that occurs when the sea spectrum is transferred from the encounter to the absolute frequency domain.

In the last twenty years, this topic has been widely investigated throughout the world, and several advancements have been made to improve the analysis of ship motions and the assessment of sea state parameters. In particular, most of these research activities have been devoted to accounting for the above-mentioned 1-to-3 multivalued problem, and to developing reliable strategies to reconstruct the sea spectrum from the encounter to the absolute frequency domain. Besides this, additional efforts have also been devoted to improving the effectiveness of the spectral reconstruction methods, given that the hydrodynamic modeling of a ship advancing in a seaway provides some challenging issues that may lead to errors in the assessment of the main sea state parameters, among which are the significant wave height and period. In this respect, Pascoal et al. [156] estimated the wave spectra from the frequency analysis of a reference 59 m offshore supply vessel at zero and low advance speed by minimizing the cost functional based on the sequential quadratic programming and genetic algorithm techniques. Pascoal and Soares [157] proposed a fast iterative procedure for the assessment of directional wave spectra, based on Kalman filtering, and applied it to a typical 70 m long vessel. Nielsen and Stredulinsky [158] analyzed a set of full-scale motion measurements, obtained during the sea trials conducted on the research vessel CFAV Quest, and compared the estimated sea state conditions against the relevant ones gathered by the wave radar processor Wave Monitoring System (WaMoS-II), installed on the considered vessel. Montazeri et al. [159] developed and applied a simplified parametric approach to estimate the wave parameters, based on the spectral moments of resembled sea spectra and a partitioning method to separately estimate the wind and swell components. Nielsen [160] developed an improved method to transform the wave spectrum from the encounter to the absolute frequency domain, consisting of two pseudo-algorithms, the former based on the spectral moments of the resembled spectra, the latter consisting of an optimization method, applied to a class of parametric spectra. Brodtkorb et al. [161] developed an online sea state assessment algorithm, based on the analysis of heave, roll and pitch motions, with no a priori assumptions related to the wave spectrum shape. The algorithm was implemented in a dynamic positioning model and tested through real simulations under different sea state conditions. Piscopo et al. [162] developed a new wave spectrum reconstruction procedure, based on the combined analysis of heave and pitch motions, and tested it against a set of numerical simulations carried out on the reference S175 containership, with different sea state conditions, speeds and heading angles. Nielsen and Diez [163] compared the estimated sea state conditions, gathered from an in-service containership, with the relevant values obtained from a hindcast study, and discussed some aspects concerning the effect of the vessel speed on the reliability of the measured data. Finally, Pennino et al. [164] applied a parametric wave spectrum resembling procedure to a set of real motion measurements, taken onboard the research vessel “Laura Bassi” during an oceanographic campaign in the Antarctic Ocean carried out during January and February 2020, and compared them against a set of weather forecast data provided by the global-WAM model.

### 5.2. Methodology

The employment of a vessel as a complex system capable of measuring sea state conditions is encouraged by the variety of sensors and recording instruments, commonly installed onboard modern ships, and capable of measuring the motion and accelerations at specific points. In this respect, any vessel can be regarded as a “wave buoy” that, by means of a proper hydrodynamic model, can be employed to detect the main sea state parameters, namely, the significant wave height, the wave peak period and the spectral shape, according to the flow-chart depicted in Figure 9, which is mainly based on three subsequent steps: (i) onboard measurement of ship motions and accelerations; (ii) data analysis according to the wave buoy analogy by frequency or time-domain methods, and (iii) assessment of main sea state parameters by parametric or non-parametric models. Nevertheless, the wave buoy analogy still provides some challenging issues, mainly related to the complex hull forms that make the hydrodynamic modeling of a ship advancing in a seaway difficult, particularly when compared to a wave buoy, whose geometry is markedly simpler. Furthermore, the ship advances in a seaway at a certain speed, which implies that the Doppler shift effect needs to considered. These topics will be outlined in the following.

#### 5.2.1. Onboard Measurement

Ship motions and accelerations can be assessed by a variety of sensors, installed onboard the ship, or even by low-cost measurement systems, such as common smartphones [164], which are generally equipped with several built-in sensors that provide raw data at high sampling rates (i.e., motion sensors, accelerometers and gyroscopes). These datasets can be analyzed separately or jointly in order to provide the time series of heave, pitch and roll motions to be further analyzed by means of the wave buoy analogy.

#### 5.2.2. Data Analysis

The analysis of ship motions and accelerations is generally based on the linear assumption between the amplitude of the incoming waves and the ship’s responses, which allows for the employment of the ship complex transfer functions and the relevant Response Amplitude Operators (RAOs) [165]. In this respect, it should be remembered that the ship RAO depends on the frequency-dependent added mass, radiation damping and restoring force, which are generally assessed via boundary-elements methods. Hence, if the ship response is assumed to be linearly dependent on the height of an incoming wave with encounter circular frequency $\omega_{e}$, the amplitude of the ship motion response depends on the RAO value at the same frequency. This assumption is generally true for mild to moderate wave climates, while some errors arise in harsh weather conditions, as some nonlinearities occur. The measured data are analyzed by frequency or time domain models.

Most past research activities have been based on the employment of frequency domain models [166,167], such that the measured ship motion spectra in the encountered frequency domain are directly embodied to resemble the wave spectrum in the absolute frequency domain by means of the ship RAOs, as depicted in Figure 10. This method is based on some additional assumptions [165], namely: (i) ocean waves and ship motions are ergodic random processes, so they can be regarded as stationary, in a stochastic sense, over a sufficiently long period; (ii) ship speed and course are constant over the measurement period. These assumptions allow the employment of the standard fast Fourier transformation (FFT) method for the assessment of the ship motion spectra in the encountered frequency domain.

As regards the time domain models, most of the research in this field has been carried out in the last decade alone. In this case, the assessment of sea state parameters is performed after solving the ship motion equations in the time domain by means of a proper algorithm, such as the one developed by Pascoal and Soares [157] and based on Kalman filtering, after introducing the wave in phase and quadrature components as state variables. A typical flow-chart of the sea state assessment by time-domain analysis is depicted in Figure 11.

#### 5.2.3. Assessment of Sea State Parameters

After detecting the wave spectrum or the elevation time history, the assessment of the main sea state parameters, namely, the significant wave height, the wave peak period and the spectral shape in the absolute frequency domain, can be performed by parametric or non-parametric models. The former are based on the detection of the unknown parameters of a given analytical spectrum, such as Bretschneider or JONSWAP, while the latter assumes a non-negative constraint on the spectral amplitude if the spectral shape is not specified a priori, based on the equivalence of the 0th-order spectral moment. Obviously, when the sea spectrum is resembled from the encounter $\omega_{e}$ to the absolute ω frequency domain, the following equation holds:(3)$$\omega_{e}=\omega -\omega^{2}\psi ,$$
with ψ=U/gcosμ denoting a nondimensional parameter, depending on the ship speed U and the heading angle μ between the ship route and the prevailing wave direction, which, in turn, equal 180 and 0 deg for the head and following seas, respectively. The dependence between the encounter and the absolute frequencies is uniquely determined when μ ≥ 90 deg, i.e., ψ ≤ 0, while the well-known 1-to-3 multivalued problem occurs when μ < 90 deg, i.e., ψ > 0, as depicted in Figure 12.

In the latter case, some issues arise when the sea spectrum is transformed from the encounter to the absolute frequency domain, provided that, when $\omega_{e}$ is less than 1/4ψ, three absolute frequencies are detected, so that the amplitude of the wave spectrum and the absolute wave frequencies are not uniquely determined. This issue can be managed by some approximate techniques [165].

### 5.3. Future Improvements

The concept of the “wave buoy analogy” is still not widely used in practice, even though it is mature enough to be comparable with other well-stablished technologies, such as the wave radars that are commonly installed onboard modern ships. In this respect, the assessment of sea state parameters by the measurement and analysis of ship motions can be considered as complementary to all the other technologies, even if some weak points need to be further improved, such as the following:The analysis of nonstationary data that may compromise the accuracy and reliability of sea state estimates;The selection of the most suitable ship motions to be endorsed in the assessment of sea state parameters, depending on the ship’s operational conditions;The employment of different types of sensors to improve the reliability of the measurement system, based on sensor fusion techniques.

The solution of these main issues will be helpful in developing a network of ships acting as wave buoys, and in enlarging the wave data and statistics that are available throughout the world.

## 6. Measurement Based on Microseism Observations

### 6.1. Microseism

Nowadays, seismologists are able to derive rich information via the study of signals that, until a couple of decades ago, were considered to be just noise, such as the so-called microseism. This is the most continuous and ubiquitous seismic signal on Earth, and is mostly generated by the ocean–solid Earth interaction [168,169,170]. On the basis of its source mechanism and spectral content, microseism is generally classified as primary, secondary, or short period secondary. Primary microseism (hereafter referred to as PM), also called “single-frequency” microseism, shows the same spectral content as the ocean waves (period 13–20 s), and its source is associated with the energy transfer of ocean waves breaking/shoaling against the shoreline [169,170]. Secondary microseism (SM), also known as “double-frequency” microseism, has shorter periods (5–10 s) and higher amplitudes than primary microseism, and it is likely to be generated by interactions between waves of the same frequency traveling in opposite directions [168,170,171]. Short period secondary microseism (SPSM) is characterized by a period shorter than 5 s, and its source mechanism is generally linked to local nearshore wave–wave interactions, influenced by local winds [172,173].

Microseism amplitudes are characterized by clear seasonal modulations. At temperate latitudes, microseism amplitudes reach their maximum values during the winter seasons, when the oceans are stormier, and their lowest values during the summer seasons [174]. However, such a modulation is different between low- and high-latitude areas: in the former, noise amplitude is mostly stable over the year, that could be because of the negligible seasonal changes close to the equator; in the latter, the modulation pattern is different, as the sea ice drastically reduces the energy transfer from the ocean to the solid Earth [174,175,176,177,178,179]. Figure 13 shows a spectrogram of the vertical component of the seismic signal recorded at an EPOZ station (installed close to the coastline of Eastern Sicily, Italy) in 2010–2017, as well as the median spectrum of the spectra composing the spectrogram. The median spectrum is characterized by peaks falling roughly within the bands SPSM, SM and PM. The spectrogram illustrates the amplitude’s seasonal modulation, as expected at temperate latitudes, with maxima during the winter seasons and minima during the summers.

### 6.2. Applications

Due to its features (primarily, continuity over time and space), microseism investigations have broad applications, such as the imaging of the crust and upper mantle by seismic noise tomography [181,182], and the detection of seismic velocity changes in both tectonic and volcanic areas [183,184].

In addition, the aforementioned source mechanism make microseism useful to inferring climate changes [174,175,185]. In particular, Grevemeyer et al. [185] used a 40-year-long record of wintertime microseism (1954–1998) to reconstruct the wave climate in the northeast Atlantic Ocean. They detected an intensification in the occurrence rate of strong microseism activity, due to the increase in the sea wave height that took place in northeast Atlantic Ocean in the second half of the analyzed period.

Finally, several authors have explored the ability of microseism analysis to provide quantitative information on the sea state, mainly in terms of sea wave height [171,186,187,188,189,190,191,192]. Bromirski et al. [187] analyzed buoy and seismometer data collected during the period 1997–1998 in California and derived site-specific seismic-to-wave transfer functions. Ferretti et al. [191] defined an automatic procedure, based mainly on a set of empirical relations [189,190], for estimating significant sea wave heights in near real-time from the spectral energy content of the microseism, and presented an application in the Ligurian Sea (Italy). Other authors have applied a more physics-based approach to quantitatively link microseism and sea wave height [170,171,193,194]. Cannata et al. [180] analyzed the microseism recorded by six seismic stations located close to the Eastern Sicilian coastline, and proposed a machine learning-based approach to calculate a regression model for reconstructing the temporal and spatial variation in sea wave height using the microseism recorded at multiple seismic stations in different frequency bands.

Despite the good correlation found in these papers between microseism amplitude and sea wave height, there are limitations [34,195]. First, the relation between these phenomena can be time-varying. Indeed, in the case of secondary microseism, if the opposing waves are generated by coastal reflection, a particular relation is valid. However, when the opposing waves are due to two uncorrelated wave systems, that relation is not valid anymore [34]. In addition, the locations of the microseism sources can be difficult to determine, and so the wave parameters in a given location in the sea can be difficult to define. This second issue can be solved using multiple sparse stations [192,196] or station arrays (that is, a certain number of seismic stations placed at discrete points in a well-defined configuration [197,198]).

### 6.3. Advantages of Microseism Monitoring

Microseism can be a valid complementary tool for monitoring sea wave activity. Indeed, although the microseism does not contain direct information on the sea state, and there could be limitations in deriving sea wave data from the microseism (see Section 6.2), there are several advantages. First of all, seismic stations have lower costs of both installation and maintenance compared to other instruments routinely used for sea wave monitoring. In addition, microseism is recorded continuously with a sampling frequency from tens to hundreds of Hz and is acquired at a very high temporal resolution. The spatial resolution depends on the number of stations installed close to the coastline [180]. Furthermore, in most areas, it is not necessary to install a seismic network specifically to record the microseism and then for sea wave height monitoring, but it is possible to use the seismic stations installed to monitor seismic and volcanic activities. In particular, broadband seismometers (that is, seismometers able to detect very weak ground motions over a wide frequency band, which extends from tens of seconds to tens of Hz) installed worldwide are very useful for microseism studies, as they record the whole microseism band [175,176]. Finally, as seismometers were among the first geophysical instruments to be installed on Earth, one of the greatest areas of interest regards the possibility of deriving very long time series of microseism amplitude, which could highlight the long term variability in sea wave activity [34,184].

## 7. Networks for Sea Wave Monitoring

### 7.1. The Added Value of Networking for Marine Weather Forecasting

The variability in sea wave conditions can greatly affect marine operations, off-shore and coastal infrastructures, and environmental aspects. This may be relevant to climatic patterns associated with large-scale currents, and to the parameterization of air–sea flows. For all these reasons, the required spatial and temporal resolutions necessitate extended networks of homogeneous sensors. Long time series of systematic and simultaneous observations (at the surface, in the water column, or at stations onshore), carried out with the goal of collecting wave data at sufficient spatial and temporal densities, are crucial to ocean exploitation.

For hundreds of years, collections of visual estimates of sea waves have been used to derive wave statistics, based on the appearance of the surface of the sea with reference to the Beaufort scale. These have been suitable for use by marine officers and for producing climate summaries on a global scale, such as the U.S. Navy Marine Climatic Atlas of the World or the UK Marine Data Bank. These atlases of ocean weather observations have long been the only global source of observed data, and are also used for design purposes. Starting from this information, it was also possible to organize continuing and readily accessible archives of global climate information, such as the International Comprehensive Ocean Atmosphere Data Set (ICOADS) [199,200]. Nowadays, numerical modeling and satellite missions—the main sources of wave information at the global scale [201]—are combined with in situ data for calibration and validation [202,203,204]. To meet these needs, the design and maintenance of real-time sea wave monitoring networks is an essential activity in any monitoring program. The observations derived from moored buoys are considered to be better quality than ship observations with regard to the accuracy and reliability of measurements [205,206]. Other techniques are presented in this paper. It is clear that, in order to effectively describe the state of the sea, a system is needed that integrates in situ and remote observations with numerical modeling in order to achieve the spatial and temporal resolution that the description of such a complex phenomenon requires. In fact, if the numerical models can meet the need for continuous data over time and with a high spatial resolution, a fundamental aspect in the development of a model is the verification of the reliability of the predictions of the model itself, because this outlines the behavior of the model in different meteorological situations, highlighting its systematic characteristics and assessing its reliability, under both average and extreme conditions, long periods and in the current situation. A good verification system allows one to not only understand where to intervene to make improvements, but above all, it allows for the best use to be made of the fields provided by the model according to the forecast objective.

### 7.2. Standards for Design and Management of a Sea-Waves Monitoring Network

There are specific criteria for the design of a network, depending on the purpose of the monitoring, related to specialized operators who conduct monitoring activities and possess specific training and experience. Generally, most sea wave networks are managed by different governmental entities in charge of national weather programs or environmental monitoring and protection programs; research centers mainly contribute to sampling and data analysis. Observation systems are also used to provide commercial services, which are sometimes dedicated to very restricted groups of users and are generally not composed of an observational network. The shared and integrated management of existing observing systems would be preferable, in order to obtain a more efficient monitoring system.

The basic guidelines for designing meteorological observation networks, including marine wave monitoring networks, are provided by the WMO, the United Nations specialized agency for international cooperation and coordination in the observation of the Earth’s atmosphere and its interactions with the land, oceans, weather, and climate it produces [207]. The objective of the WMO is to support the meteorological activities of the Member States through the definition of good practices in the field of meteorological observations, to promote an adequate level of uniformity and standardization in the practices and procedures used for the measurements, and to facilitate cooperation in observations. The WMO Integrated Global Observing System (WIGOS) addresses the observation needs of the WMO, including the main areas of the standardization of observation tools and methods, related metadata, and file formats. It provides an overview of all operating systems for the matrixes associated with the Earth’s atmosphere; moreover, it contributes to the co-sponsored observation systems supporting the activities of the WMO [208,209]. The observations collected are subjected to quality controls according to the technical standards defined by the WMO Instruments and Methods of Observation Program (IMOP) and subsequently made public through the WMO Information System (WIS).

The Intergovernmental Oceanographic Commission of UNESCO (IOC) is the organization responsible for marine science within the United Nations. The IOC allows the Member States to coordinate ocean research and services, as well as related activities regarding oceanographic measurements focused on sustained ocean observing and data management activities, encompassed in the Global Ocean Observing System (GOOS) and carrying out regional activities through GOOS Regional Alliances [210], the observing program area of the Joint WMO-IOC Technical Commission for Oceanography and Marine Meteorology (JCOMM), and the International Oceanographic Data and Information Exchange (IODE). The implementation of the international program is carried out by the Member States through their operational structures, such as government agencies, navies, and oceanographic research institutes. Several global thematic groups, observation networks, and regional alliances have been established with the agreement of member states to work together and ensure long-term sustainable ocean observation. Within JCOMM, standards, procedures, and recommendations have been developed to provide all marine data and product users with an integrated marine observation data management service system based on cutting-edge technologies to meet changing global needs [211]. The JCOMM has also set itself the ambitious goal of coordinating the long-term updating and maintenance of an integrated system for the observation and management of global marine, meteorological and oceanographic data.

The WMO guidelines state that the identification of the best equipment should take into account the monitored parameters, the location of the station, and the purpose of the monitoring activity. The station spacing and interval between observations should correspond with the desired spatial and temporal resolution of the meteorological variables to be measured or observed. The location of each station should be representative of the conditions in space and time. The total number of stations should, for the sake of financial cost, be as small as possible but as large as necessary to meet scientific requirements. The success of a directional wave measurement network largely depends on the use of reliable and effective instrumentation that can be operated from the sea surface, across the water column, from the seafloor, or remotely. Moored and drifting buoys, ships, and stationary platforms can support complex payloads, allowing the co-located measurement of many of the GOOS Essential Ocean Variables (EOV), and they are relatively easy to upgrade and equip with additional sensors [212].

Figure 14 shows the NOAA Observing System Monitoring Center (OSMC) dashboard in support of the goals of JCOMM, taking into account the global observational capability of sea waves over the first nine months of 2021. This dashboard provides a point of access into the integrated data streams and attendant metadata for the continuous global ocean observation efforts, derived from sea wave global networks (DBCP, ARGO, SOT/VOS, GLOSS) and distributing wave data from the Global Telecommunication System (GTS).

### 7.3. Globally Integrated Sea-Waves Monitoring Networks

Observing sea waves from drifting and moored buoys has a long history. Buoy projects are now mature, and these technologies benefit from decades of field experience and international collaboration. The Data Buoy Cooperation Panel (DBCP) maintains a huge catalog of wave buoy programs that supply data for operational and research purposes. The DBCP operates through the following action groups (and buoy programs): Global Drifter Program (GDP), Tropical Moored Buoy Implementation Panel (TAO/TRITON, PIRATA, RAMA), European EUCOS Surface Marine Programme (E-SURFMAR), International Arctic Buoy Programme (IABP), International South Atlantic Buoy Programme (ISABP), North Pacific Data Buoy Advisory Panel (NPDBAP), International Buoy Program for the Indian Ocean (IBPIO), and International Programme for Antarctic Buoys (IPAB). The USA, Canada, Australia, Japan, and European countries are the major participants, but other countries with great potential to make contributions, such as India and China, are now joining in the cooperation.

The Lagrangian drifters deployed by GDP have been observing essential climate variables (ECV) since 1979. The objective of GDP is to maintain a global array of surface drifting buoys (satellite-tracked) to meet the need for an accurate and dense set of in situ observations on a global scale. These EOV and ECV are essentially the near-surface currents, sea surface temperature, sea-level atmospheric pressure, winds, salinity, and waves [6]. The first generation of the GDP, the Surface Velocity Program (SVP), played an important role in the TOGA program. The SVP drifters were the simplest version of drifters, and can only measure two parameters: ocean currents and sea surface temperature. With additional sensors, such as barometers, sonic anemometers, conductometers, and GPS, the performance of the drifters can be enhanced further to adapt to different applications.

The global tropical moored buoy arrays (TAO/TRITON, PIRATA, RAMA) are used to monitor large-scale phenomena, such as El Niño and the Southern Oscillation (ENSO), showing the importance of the annual variability in the global climate. They are deployed at depths of up to 6000 m. Measurements taken by the mooring include surface variables, as well as subsurface temperatures down to a depth of 500 m [213]. The Tropical Atmosphere Ocean (TAO) and the Triangle Trans-Ocean Buoy Network the Pacific Ocean (TRITON) mooring array were successfully fielded to span the Pacific Ocean, with the immediate ability to forecast El Nino phenomena up to a year ahead of their peak [214]. TAO/TRITON was built over the ten-year period 1985–1994, and is presently supported by the NOAA National Data Buoy Center and the Japan Agency for Marine Earth Science and Technology. The Prediction and Research Moored Array in the Tropical Atlantic (PIRATA) was designed to study ocean–atmosphere interactions in the tropical Atlantic that affect regional weather and climate variability on seasonal, interannual, and longer time scales, and is motivated by the goal of understanding and better predicting certain phenomena, such as tropical Atlantic interannual to decadal variability and climate change [215]. It was first established in the mid-1990s, and is supported by France (Meteo-France, CNRS, IFREMER), Brazil (INPE, DHN), and the USA (NOAA). The Research Moored Array for African–Asian–Australian Monsoon Analysis and Prediction (RAMA) was designed to study the Indian Ocean’s role in monsoons. The array was initiated in 2004 and has since grown through the formation of new partnerships that currently include Indonesia, China, the USA, and the Bay of Bengal Large Marine Ecosystem (BOBLME) program.

The European EUCOS Surface Marine Programme (E-SURFMAR) is an optional program involving seventeen European meteorological services (Belgium, Croatia, Cyprus, Denmark, Finland, France, Germany, Greece, Iceland, Ireland, Italy, The Netherlands, Norway, Portugal, Spain, Sweden, and the United Kingdom) that coordinates, optimizes and progressively integrates the European meteorological services’ activities in surface observations over the sea, including drifting and moored buoys, and voluntary observational ships. The program is responsible for the coordination of buoy activities carried out by the European meteorological services, and supports a Data Buoy Manager (DBM) to manage these activities [216,217]. In recent years, sea wave data from European research and operational centers have also been made available through the Copernicus Marine (CMEMS) and EMODnet data portals, with increasing adherence to the emerging FAIR Principles for data publication.

The International Programme for Antarctic Buoys (IPAB) maintains collaboration among national groups and international programs deploying buoys on the sea ice and a network of drifting buoys in the Southern Ocean.

The implementation of new assets must take into account the limits and potential of each sea wave collection methodology in relation to the different purposes to which a systematic monitoring activity can be oriented, as well as the essential activities of the maintenance, continuous testing, and evaluation of wave measurement systems. Most importantly, for a globally integrated ocean observation system, the necessary consensus has now been reached in the community of ocean observers to establish, grow and coordinate internationally [218]. For regional (or national) observation systems in the coastal sector, the network requirements are considerably more demanding, because, while the range of measurement technologies is the same, the density of the stations and their emphasis will be greater [219,220].

The development of such integrated systems, which is crucial for the proper management and safety of both coastal areas and the open sea, must take into account a series of technological and scientific advances that have broadened the spectrum of platforms available to the scientific community and institutional stakeholders. The different methodologies available must not be considered as in competition, but instead be thought of as connected in a complementary way so as to be able to take full advantage of the strengths of each. The goal is therefore the integration of all observations in order to monitor the essential parameters needed for a deeper understanding of the marine environment in the short and long term.

## 8. Conclusions

Since sea waves have a great impact on human activities, accurate measurement and monitoring at different geographical scales is a major goal at the international, national and local levels. There are several measurement techniques available, with different degrees of development and with, typically, complementary features. An ample selection of such techniques, whose application is essential to meteorology, coastal safety, navigation, and renewable energy derived from the sea, has been here reviewed and includes buoys, satellite observation, coastal radars, shipboard observation, and microseism analysis. After a brief presentation of the measurement principle, the degree of development has been outlined, and future needs and trends have been prospected. Furthermore, it has been stressed that the present and the future of ocean measurement lie in observation networks that complement, and interact with, each other. In order to effectively describe the state of the sea, a system is needed that integrates in situ and remote observations with numerical modeling in order to achieve the spatial and temporal resolution that the description of such a complex phenomenon requires.

## Figures and Tables

**Figure 1 sensors-22-00078-f001:**
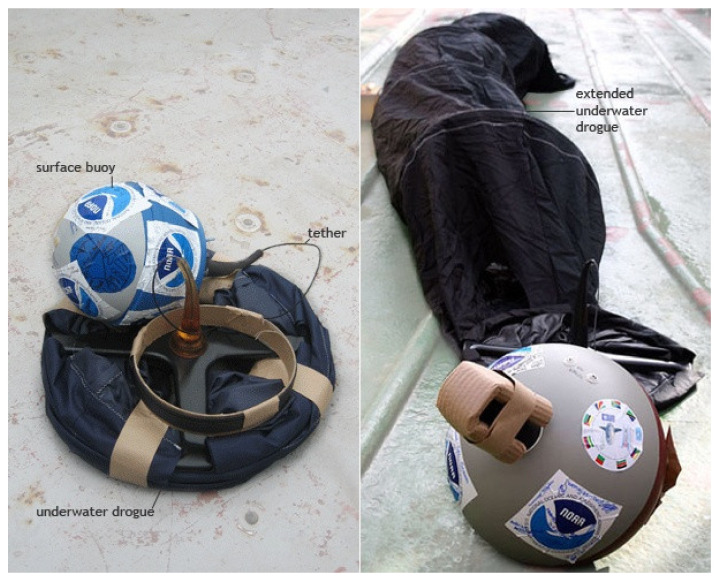
A drifter neatly compressed for deployment (**left**) and with the nylon drogue fully extended (**right**), as it will be in the water once the cardboard wraps dissolve. Photos courtesy of National Oceanic and Atmospheric Administration (NOAA)—Atlantic Oceanographic and Meteorological Laboratory.

**Figure 2 sensors-22-00078-f002:**
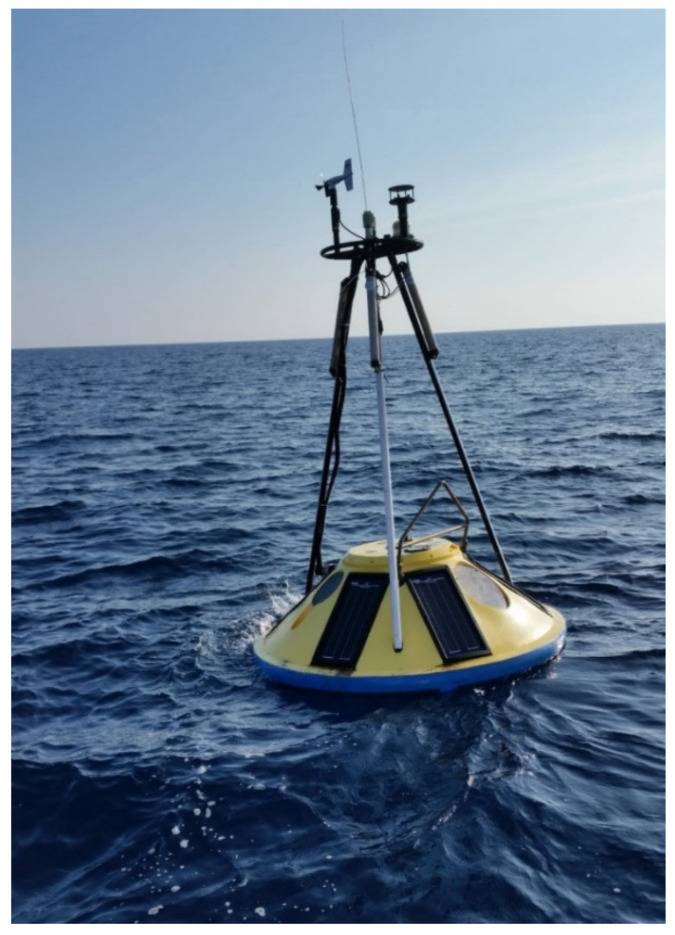
The Capo Mele moored buoy is a Fugro Oceanor SEAWATCH Midi 185 (radius of 1.85 m). It is located at about 3NM from Andora port (Liguria, Italy). Thanks to a weather station 2 m above sea level, the buoy measures wind (direction, intensity, gust), atmospheric pressure, humidity, and air temperature. The observed marine parameters are wave significant height and maximum peak, wave period and direction, current intensity and direction from 3.5 to 70 m, and sea surface temperature. Photo courtesy of Regional Agency for Environmental Protection of Liguria Region (ARPAL).

**Figure 3 sensors-22-00078-f003:**
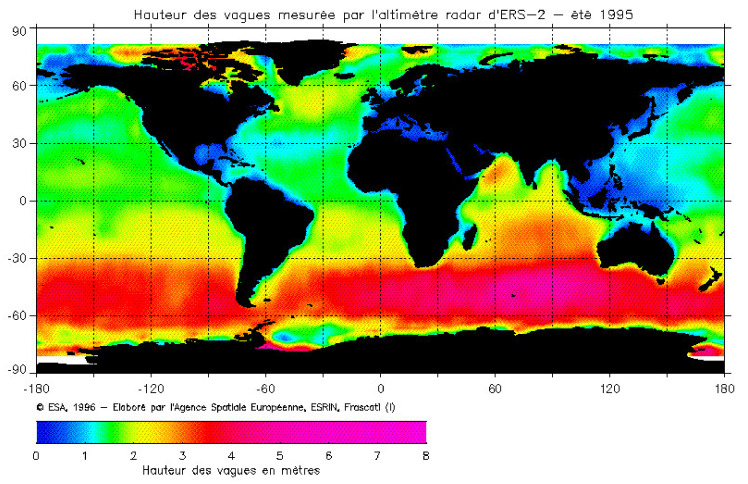
SWH global mapping. Courtesy of ESA. https://www.esa.int/ESA_Multimedia/Images/2002/03/Significant_Wave_Height_Measured_by_the_ERS_Radar_Altimeter#.YZvksOv7zJs.link (access on 21 December 2021).

**Figure 4 sensors-22-00078-f004:**
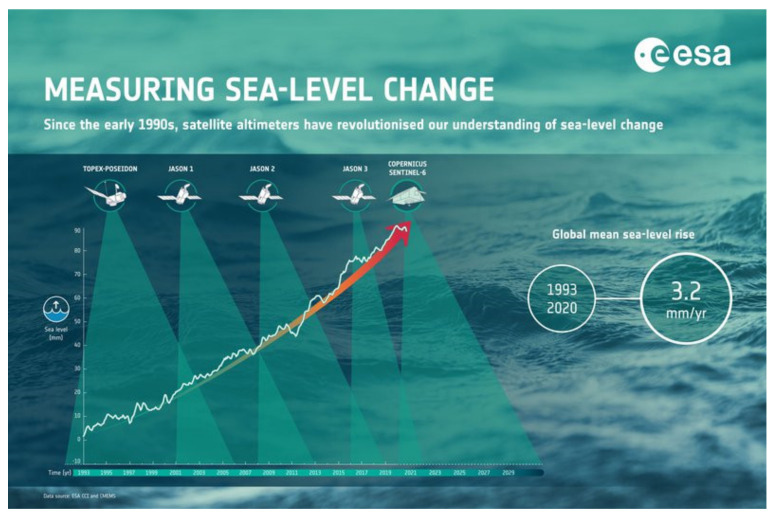
Global mean sea level (seasonal signals removed). Climate Change Initiative. Courtesy of ESA. https://climate.esa.int/en/news-events/coastal-observations-boosted-new-reference-satellite/ (accessed on 21 December 2021).

**Figure 5 sensors-22-00078-f005:**
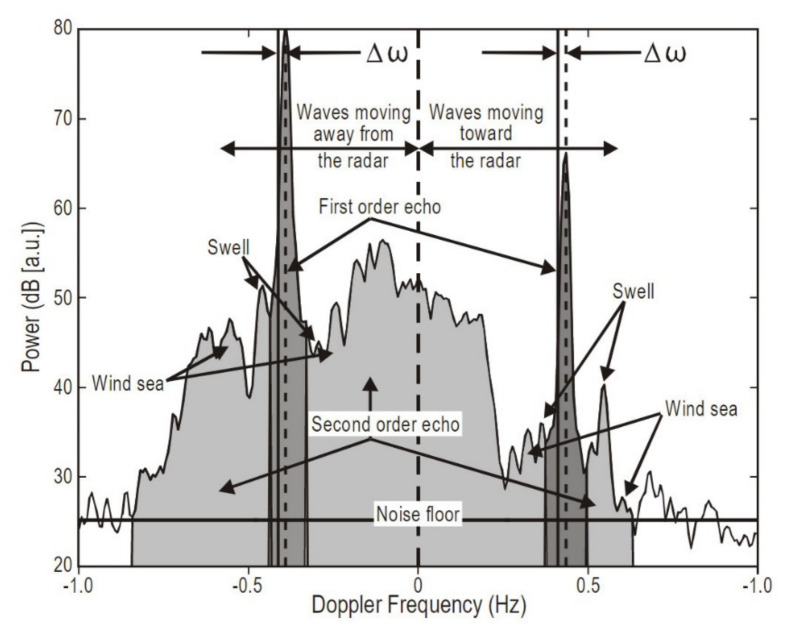
Main characteristics of a typical HF radar spectrum [114].

**Figure 6 sensors-22-00078-f006:**
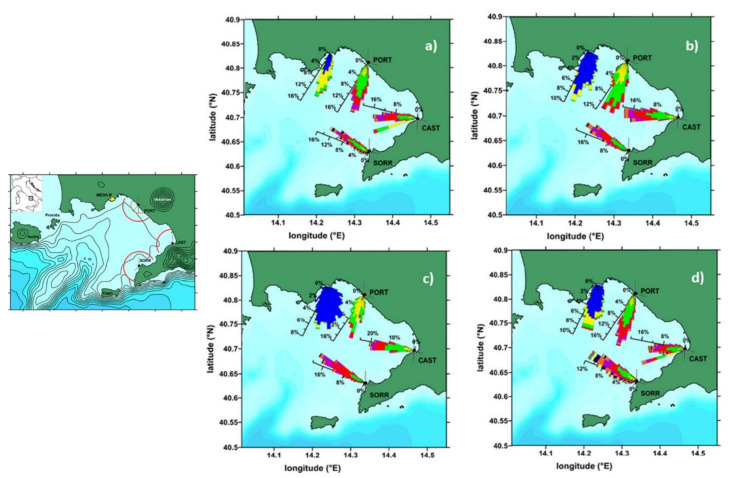
Seasonal rose diagrams for the waves measured between May 2008 and December 2012 by three HF radar stations installed in the Gulf of Naples, and between November 2015 and December 2018 by an ADCP mounted on a MEDA elastic beacon: (**a**) winter; (**b**) spring; (**c**) summer; and (**d**) autumn (the maps show the locations of the different instruments and the extension of the range cell utilized around the radar antennas). Adapted from [127].

**Figure 7 sensors-22-00078-f007:**
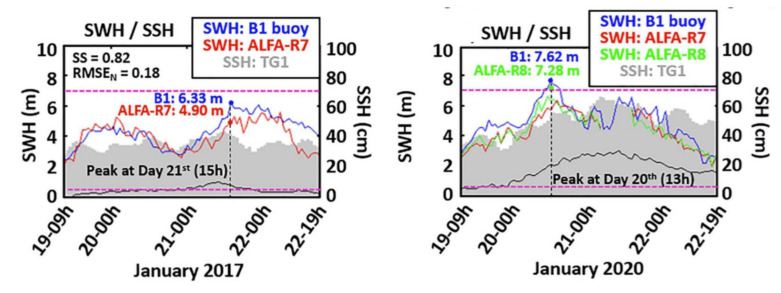
Comparison of measurements collected by different instruments during two storms that occurred in 2017 and 2020 (adapted from [121]). The abbreviations in the plots are explained in the main text of this paper.

**Figure 8 sensors-22-00078-f008:**
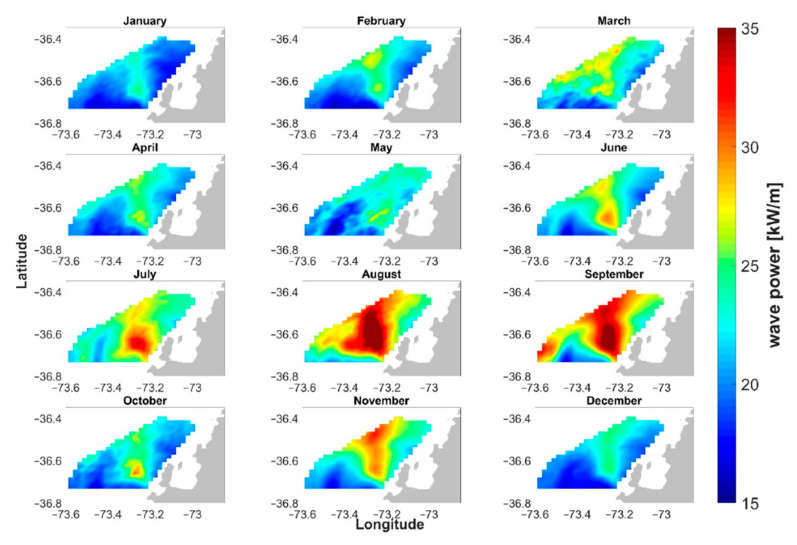
Wave potential monthly mean fields estimated on the basis of data provided by a phased array HF radar system installed off the coast of Chile [140].

**Figure 9 sensors-22-00078-f009:**
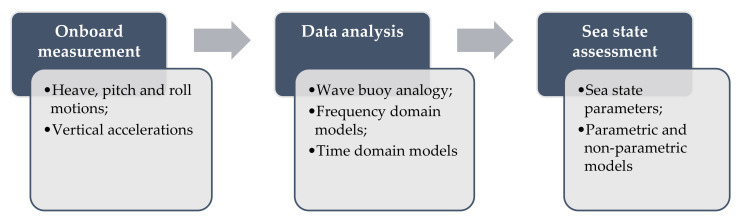
Typical flow chart of sea state onboard measurements.

**Figure 10 sensors-22-00078-f010:**

Sea state assessment by frequency domain analysis.

**Figure 11 sensors-22-00078-f011:**

Sea state assessment by time domain analysis.

**Figure 12 sensors-22-00078-f012:**
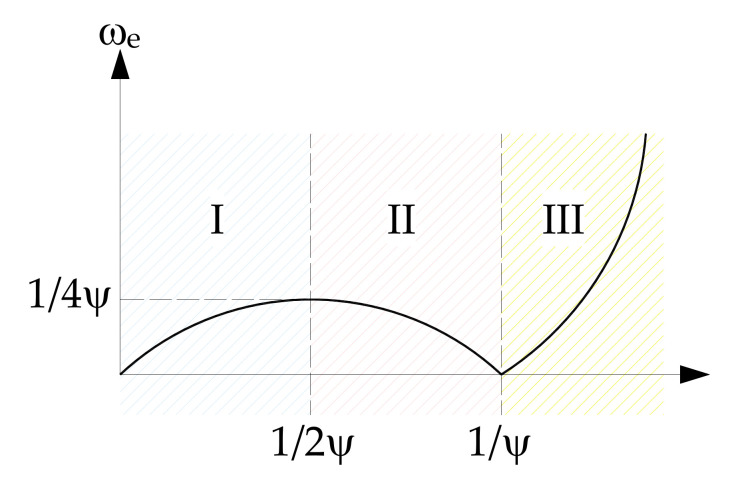
The 1-to-3 multivalued problem between the encounter and the absolute wave frequencies.

**Figure 13 sensors-22-00078-f013:**
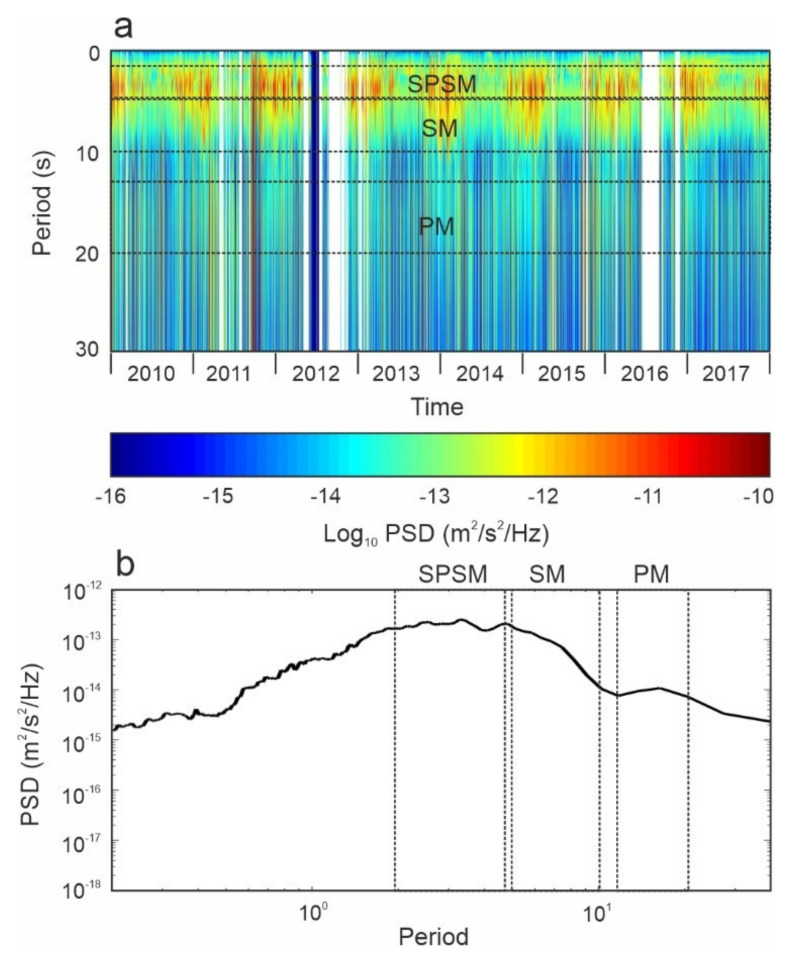
(**a**) Spectrogram of the vertical component of the seismic signal recorded by the EPOZ station (installed close to the coastline of Eastern Sicily, Italy) in 2010–2017. (**b**) Median spectrum of the spectra composing the spectrogram in (**a**). SPSM, SM, and PM and the corresponding dashed rectangles indicate the frequency bands characterized by short period secondary microseism, secondary microseism, and primary microseism, respectively. The acronym PSD in both color bar of (**a**) and *y*-axis of (**b**) indicates power spectral density (modified from [180]).

**Figure 14 sensors-22-00078-f014:**
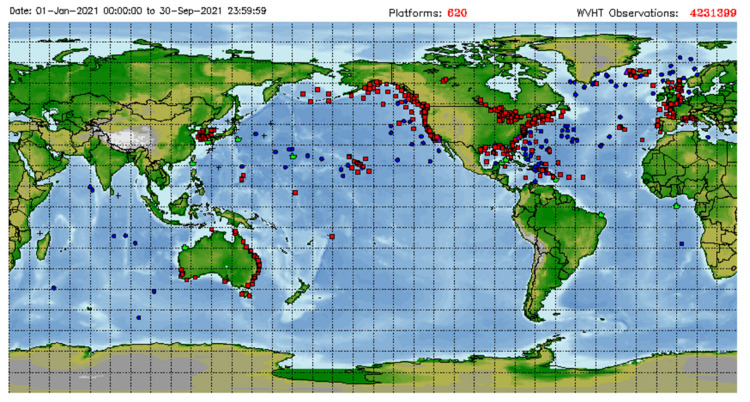
Global sea wave monitoring networks reporting on GTS during the first nine months of 2021; drifting buoys (blue dots), moored buoys (red squares), ships (green icons), others (black crosses). Map generated by: http://osmc.noaa.gov/Monitor/OSMC/OSMC.html (accessed on 29 October 2021).

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
