# Peer review of "Measurement of Sea Waves"

_sensors, 2021, doi:10.3390/s22010078_

Round 1

Reviewer 1 Report

This paper highlights the importance of accurate, large-scale, long-term and high frequency acquisition of wave information to human activities, and comprehensively summarizes the existing wave observation methods, including buoy measurement, satellite microwave remote sensing, HF radar measurement, shipboard sea state observation, microseism observation and emphasizes the importance of the global wave observation network, It also lists the organizations and projects that have contributed to the collection, sharing and standard setting of global wave information, and puts forward valuable suggestions for the construction of global wave observation network in the future.

Buoys can be divided into drifting buoys and moored buoys. The moored buoys is divided into water particles-following type and surface-following type. It is mentioned that the new buoys uses GPS to obtain wave information. The paper will be enriched if the advantages and disadvantages of the new buoys using GPS and the traditional buoys using accelerometer are further compared from the measurement accuracy and application range.

Spaceborne microwave radar has the ability to observe the ocean all day, all weather and in a wide range. Synthetic aperture radar has made great achievements in land imaging applications, but it faces many difficulties in the application of random moving ocean surface. The third part of the paper introduces the application of scatterometer, radiometer and synthetic aperture radar in ocean wind field inversion, and also briefly mentions the role of altimeter and synthetic aperture radar in ocean wave information extraction. In keeping with the title" Measurement of Sea Waves ", it would be better to focus more on ocean wave remote sensing. Such as briefly introduce the principle of obtaining significant wave height by altimeter and the modulation of ocean waves on SAR images, lists several classical methods of ocean wave spectrum inversion based on SAR data, and analyzes their advantages, disadvantages and the scope of application.

Some newly published literature on ocean wave parameter extraction can be introduced in this paper. The application of cut-off wavelength in sea surface wind field inversion is mentioned, and in recent years, cut-off wavelength is also used in the inversion of significant wave height and wave orbital velocity. After the method of using the fully polarized image to eliminate the hydrodynamic modulation function and invert the wave slope spectrum mentioned in this paper, the nonlinear mapping of the new polarized image spectrum to the SAR image spectrum is developed, which expands the application scope of this method. The ocean wave inversion method of dual antenna interferometric synthetic aperture radar can also be briefly introduced in this paper, including along orbit interference and cross orbit interference. This method does not depend on the modulation function of real aperture radar and has certain advantages. With the development of computer science, the means of deep learning has also been applied to ocean remote sensing to improve the accuracy of wave parameter inversion.

Are there any statistical data or other methods to quantify the accuracy of the various wave measurements mentioned in this article? If so, present them in the article so that readers can feel the quality and feasibility of the measurement method more intuitively.

There are also some minor issues worth noting in the article:

In the sentence from line 318 to line 322. The linearly modulated chirp pulses traveling at the speed of light, but the pulses travel twice as far as radar to its target.

It might be better to replace‘determined by’ with ‘benefit from’ in line 321, since the resolution of SAR is determined by the aperture of the antenna.

It would be better to replace Figure 3 with a SAR image of the waves, since the Sea-surface salinity is not what we care about in this paper.

The sentence from line 318 to line 322 misuses causation,

There are also some spelling problems such as in line 694 ‘hindicast’ and line 920 ‘end’. 

Reviewer 2 Report

Different techniques and approaches in the measurement of sea waves are reviewed. Comprehensive information is provided however, it is not well classified with the main topic being lost in some sections. I suggest rewriting most of the sections while focusing on delivering the main message of each section with special attention paid to the flow of the sentences and paragraphs being placed one after another. Below are some more specific comments:

Section2: Waves buoys are divided into drifting and moored buoys. In my opinion that is the different deployment types rather than buoy types!

A lot of information about buoys is included in this section while not properly organized. Below is how this section is designed:

2. Wave buoys

2.1 Measurement techniques from buoys

2.1 Moored buoys

2.2 Moored buoys wave sensors

Firstly, there are two sections numbered “2.1” and two sections named “Moored buoys”!! In the “Measurement techniques” section, drifting buoys are described!! This improper organization of the information makes it difficult to follow the flow of information leaving readers lost!!

Lines 96-98: “These observations are used in multiple applications, including strengthening the quality and accuracy of severe and routine weather forecasting, improving coastal ocean circulation models, environmental and ecosystem monitoring, and research topics.”
Is it to say an application of wave buoy observations is to improve “research topics”?!

Line 122: “Tush”
Is it “Thus”?

Lines 123-124: “Directional wave spectra (DWS) drifters are a new generation GPS-based tracking and wave engine”
I don’t get what the authors mean here by “wave engine”!

Lines 124-125: “onboard processing the “first five” directional Fourier coefficients”
It is not clear what the authors mean by “onboard”!

Line 134: “measure currents, temperature, and salinity”
How about waves? Are waves not the main topic of this paper?

Lines 135-135: “Floats differ, however, in that they profile the deeper waters of the ocean.”
This sentence is unclear!

Lines 142-143: “Ocean trajectories, which are called a Lagrangian description of the flow…”
Do you mean “Ocean trajectories” are “a Lagrangian description of the flow”?! That doesn’t sound correct!!

Line 153: “They can sit on sea-ice or sit in the open ocean”
What does it mean for an “ice buoy” to sit in the open ocean?!

Line 181: Specify what  “WMO” stands for?

Line 182: What is defined as a “dangerous storm”?!

Lines 206-208: “The essential spectral characteristics of the waves can be represented, as a first approximation, by a standard theoretical formulation defined by a small number of independent parameters.”
It is not clear what the authors mean by this sentence, a very poor choice of words for a scientific article!

Line 208: “quite similar different models”
Similar or different?!

Lines 209-211: “These spectra represent conditions in deep water generation area and can be used to relate the shape of the spectrum of the wind waves.”
Only deep water? Only wind waves? How about swell waves? What does differentiate Pierson-Moskowitz from JONSWAP spectrum?

Lines 2011-214: “A smoothed version of the directional spectrum can be constructed by measures from the “first-5” Fourier coefficients so that the shape of the spectrum used does not give negative values, but a large enough weighting function causes a noticeable smoothing of the spectrum.”
What does it mean? Provide references.

Lines 230-231: Explain “Maximum Entropy Method” and provide references.

Section3: This is a very long section (more than 4 pages) with no subsection. The information is not classified properly with the message sounding disconnected!

Line 306: “in some areas only few in situ measurements are available”
Proved examples of such areas. For instance, the Southern Ocean.

Line 307: “less dependent on atmosphere phenomena”
What are “atmosphere phenomena” give an example.

Line 316: “due a proper SAR processing chain”
What does that mean?

Lines 322-324: “Further, due to its coherent nature, within each resolution cell, fading due to the interference of the ensemble of elementary scatterers within the cell generates, in SAR imagery, the so-called speckle.”
This sentence is not clear and needs to be rewritten!

Section 4: Radars can also be deployed on ships for wave measurement purposes. Would have been better to, more generally, explain radars in this section regardless of them being deployed in coastal regions or on the ship.

Lines 507-510: “The main peaks correspond to the first-order scatter from the so-called Bragg waves: when trains of surface ocean waves have a wavelength half as long as the transmitted signal’s one, a coherent resonance occurs, analogous to the Bragg effect in atomic lattice detection by x rays, as first observed in [96] and later clarified in [97-101].”
Long sentence with confusing punctuation!

Section5:

Section 5.1: In the listing of previous works, a bigger picture needs to be described. What led to one study after another? What was missing and what questions were researchers trying to answer?

Section 5.2: What are the limitations of wave buoy analogy? For example, ships are of more complex geometry than buoys. The vessel speeds also need to be taken into account.

Section 5.2.2: RAO deserves more explanation. How can it be calculated? Was does it depend on?

Section 7:

Line 879-882: “The variation of sea-waves conditions can greatly affect marine operations, off-shore and coastal infrastructures, environmental aspects, in addition, it can be relevant to climatic patterns associated with large scales currents and enters into the parameterization of air-sea flows.”
The sentence is not clear and the punctuations are confusing!

Lines 887-891: “For hundreds of years collections of visual estimates of sea waves based on the appearance of the surface of the sea, by reference to the Beaufort scale, were used to derive wave statistics for the use of marine officers and to produce climate summaries on a global scale, such as the U.S. Navy Marine Climatic Atlas of the World or the UK Marine Data Bank.”
Very long sentence with confusing punctuations!

Round 2

Reviewer 1 Report

The paper has been improved after modification, but there are still the following problems.

1.The first paragraph of section 3.1 (lines 330 to 335) is not clear. Such as “providing a longer revisit time” should be replaced with “Providing all day observation”. 

2. In the section of the along-track SAR interferometry, it is suggested to add the following statement. The sea surface radial velocity can be calculated from the interference phase of each resolution unit, and then the wave spectrum is obtained from the radial velocity. However, this method is still affected by velocity bunching.

3.Cross-track SAR interferometry can also be briefly described, although it is mainly implemented on airborne platform.

Reviewer 2 Report

Thanks for incorporating the previous comments in the manuscript. It is notably improved however, I suggest paying more attention to sections 2 and 3. The information in these two sections is not well organized and the headers are confusing for the readers. My other specific comments are as follows:

Lines 53 to 56: “The wave height is usually expressed as significant wave height Hs, defined as the mean value of the highest third of wave heights in the Rayleigh distribution (wave heights in deep water are nearly Rayleigh-distributed [1]), or it can be estimated from the spectrum obtained from a time series of sea surface elevation.”
1. Replace “highest third” with “highest one-third”.
2. This sentence does not sound correct. The significant wave height is described as “the mean value of the highest [one] third of wave heights” no matter what the distribution is. I do not quite get why Rayleigh distribution deep water waves are brought up here?

Lines 68 and 69: “The sea surface can be described by the two-dimensional spectrum of incoming waves in frequency and direction…”
What do you mean by “incoming waves”?

Equation 1: All parameters need to be described. What is “n”, for example?

Lines 84 and 85: “Significant advances have been made in the measurement of the directional wave spectrum over the past decades…”
This sentence gives the impression that the paper is going to focus on “the measurement of the directional wave spectrum” which is not the case. I suggest replacing “the measurement of the directional wave spectrum” with “the measurement of waves”.

Line 330: “…represents a special observational tool to monitor phenomena not measurable in situ.”
This sentence doesn’t sound quite right. Maybe better to emphasize regions that are not easily accessible for in-situ measurements rather than the “phenomena”.

Section 2: The subsection headers are still confusing. What is the difference between “2.2 Moored buoys” and “2.3 Moored buoys wave sensors”?

Section 3: The information is not properly organized in this section. Subsections are not well chosen based on the information provided:

  1. Satellite remote sensing
    3.1 Background
    3.2 Review of the main marine added value products

    What is “marine added value products”? Why altimeters are not discussed in this sub-section?
    3.3 Sea Waves
    This is not the best sub-section header, and some information is repetitive e.g., “[altimeter] is a large-scale nadir-facing active sensor”.
    Also, it is worth adding some information about Chinese-French Oceanography Satellite (CFOSAT) in this section.

Lines 575 to 584: In my opinion, this sentencing and the information provided in the last paragraph of Section 3 do not suit an academic/scientific publication.

Line 587: “Coastal HF radars”
What does HF stand for?

Line 624: “upper threshold for accurate wave detection”
Is it “upper threshold for accurate wave [height] detection”?

Line 775: Replace “WaMos II” with “Wave Monitoring System (WaMoS-II)”.

Figure 12: Replace “Wave elevation” with “Water elevation”.

Line 896: Replace “help” with “helpful”.

Line 920: I suggest replacing “probably because of” with “that could be because of”.
